# The meaning of momentary psychotic-like experiences in a non-clinical sample: A personality perspective

Goran Knežević[1]*, Ljiljana B. Lazarević[2], Aleksandar Zorić[1]

**1** Faculty of Philosophy, Department of Psychology, University of Belgrade, Belgrade, Serbia, **2** Faculty of Philosophy, Institute of Psychology, University of Belgrade, Belgrade, Serbia

* gknezevi@f.bg.ac.rs

## Abstract

The relationships between Momentary Psychotic-Like Experiences (MPLEs) and HEXACO —complemented by the proneness to PLEs conceptualized as a basic personality trait (Disintegration), and a maladaptive trait (PID-5 Psychoticism)—were investigated in a prospective study that includes experience-sampling methodology (ESM). The main goal was to investigate whether MPLEs are better predicted by HEXACO or measures of the dispositional proneness to PLEs. A sample of 180 participants assessed MPLEs and affective states they experienced in the previous two hours, twice per day, with semi-randomly set assessment time-points, during seven days, by using ESM. Personality inventories were administered 1–2 months earlier. MPLEs were better predicted by the measures of dispositional tendencies toward PLEs than by the HEXACO, no matter whether it was broadly defined as the nine-faceted general tendency toward PLEs (Disintegration), or narrowly as three-faceted positive psychotic-like symptoms of maladaptive personality tendencies (PID-5—Psychoticism).

## Introduction

What is the real meaning of endorsing the items that assess psychotic-like experiences (PLEs) by the individuals from the general population i.e., those who are not psychotic? Do the inventory items designed to assess a variety of such experiences at the various levels of intensity and in different groups (including clinical and non-clinical groups) indeed capture the same (psychotic) phenotype? Based on the available evidence, the answer seems to be affirmative. Namely, studies conducted with nonclinical populations, using both structured clinical interviews and self-report measures, demonstrated that psychotic experiences and beliefs are common in those samples [1–5]. Factor analytic studies found dimensions of differently labeled subclinical psychotic phenotypes—those found in non-clinical populations—to be parallel to those found for psychosis [6,7]. Additional lines of the compelling evidence on the continual distribution of PLEs having the same meaning across clinical and non-clinical populations could be found elsewhere (see, for example, [8]).

**Data Availability Statement:** All relevant data files are available from the OSF repository (https://osf.io/94t6p/?view_only=211142e8fc914678a8a93360c3124eb0).

**Funding:** This study was supported by the Ministry of Education, Science and Technological Development (Serbia), project number 451-03-9/ 2021-14/ 200163.

**Competing interests:** The authors have declared that no competing interests exist.

Although the majority accepts the existence of the continuum of PLEs, some uneasiness seems to exist regarding its conceptualization as a personality trait, like Big Five or HEXACO traits. Even though some openly suggest to reconceptualize the tendency to PLEs as a personality trait separate from the other basic personality traits [9–12], the majority either understand it as something substantially different from so-called normal personality variations or—curiously—something that is already an aspect of the established taxonomy of personality variations. Thus, according to former, Psychoticism defined by the Alternative Model for Personality Disorders—AMPD [13] is a trait of maladaptive variations, not characterizing normal personality [14], or, as latter suggests, it should be conceptualized as extreme Openness (O) (e.g., [15]). Understanding PLEs as extreme O seems to gain considerable support among the scholars in the field (e.g., [16,17]), despite the persuasive evidence that variously defined and assessed domains of PLEs tend to factorially separate from the Big Five O (e.g., [10,11,18]) or HEXACO O (e.g., [19,20]). However, this evidence seems to influence proponents of the view to adjust it: their refined claim is that only so-called positive PLEs are substantially related to O, but only to the aspects of O such as fantasy-proneness and aesthetic interests (e.g., [21]), that is, those labeled Openness to experience (OE). Namely, it is suggested that the usual operationalizations of the O domain are overly broad, blending two distinct subfactors—OE, which is positively related to psychotic-like phenomena, and Intellect (I), which is negatively related. Thus, as argued by these authors, usually obtained near-zero correlations between O and psychosis measures are the consequence of merging these two aspects of O into one score [22].

### Basic personality traits and psychotic like-experiences

One of the frequently used personality models—HEXACO, proposes that personality space can be described along the six broad stable personality dimensions, i.e., Honesty/Humility (H), Emotionality (E), Extraversion (X), Agreeableness (A), Conscientiousness, and Openness (O) [23]. Empirical evidence, including a recent study showing its cross-cultural replicability [24], suggests that the HEXACO model provides a valid description of personality space and a useful account of individual differences [25].

As already stated, substantial empirical evidence shows that psychotic-like phenomena (broad spectrum of psychological phenomena captured by the trait-lake concepts such as psychosis continuum, psychosis proneness, subclinical psychosis [26], schizotypy [27], psychoticism [14], oddity [18], or apophenia—i.e., a tendency to see patterns in randomness, [15]) create a continuum from sub-clinical forms frequently present in the general population to the phenomena reflecting psychotic and schizophrenic disorders [8,26]. Recently, a model reconceptualizing psychosis-proneness as a basic personality trait was developed [11]. This model articulated psychosis proneness as a (surprisingly) broad, hierarchically organized, multidimensional behavioral disposition—a basic personality trait named Disintegration (D). It was demonstrated that the domain can be defined along the nine lower-level dimensions strongly converging to the higher-order D factor. These dimensions are: 1) Perceptual Distortions (PD) (depersonalization, derealization—including experiencing multiple identities, and Schneiderian first-rank symptoms); 2) General Cognitive/Executive Impairment (GC/EI) (dysregulation of attention, planning, memory, concentration, speech comprehension and production, motor control, blackouts, absent-mindedness, hypo-awareness of one's behavior, obsessions and compulsions); 3) Enhanced Awareness (EA) (dissociative involvement in engaging stimuli, increased cognition, synesthesia, and vivid reminiscence); 4) Apathy/Depression (A/DEP) (meaninglessness, chronic fatigue, hopelessness, helplessness, and suicidal ideation); 5) Paranoia (PAR) (suspiciousness, distrustfulness, oversensitivity, paranoid resentment, ideas of

persecution, and conspiracy); 6) Mania (MAN) (agitation, overactivity, extreme optimism, elevated mood, inflated self-esteem, and excessive grandiosity); 7) Flattened Affect (FA) (primarily capturing emotional numbing, emotional indifference towards the self and others, lack of future planning, and superficiality, not outward expression of emotion in face or voice); 8) Somatoform Dysregulations (SOD) (experiences of organ malfunction and damage, severe forms of sensory and motor conversions, insensitivity to pain, and general body numbing, not body image distortions which belong to the PD subdomain), and 9) Magical Thinking (MT) (feeling telepathic, having energetic connections with others, illogical thinking, belief in the afterlife and reincarnation, magical influences, and horoscopes). These nine factors (initially included Social Anhedonia factor was excluded from the domain because—despite having substantive loadings on the higher-order D factor—it was found to have primary loadings on low Extraversion in all our subsequent studies) were derived from a series of factor analyses of almost a thousand items tapping various forms of PLEs that were administered to a sample of almost 3000 participants. The remaining nine dimensions form a strong factor separate from the FFM (Five-Factor Model, [28]). The separateness of D from FFM traits was replicated across informants (self-, mother's, and father's reports), samples (undergraduate students and the general population), and units of analyses (facets and items). The separateness of D from HEXACO was demonstrated recently across three national samples [20]. Moreover, D has a normal distribution in the general population when based on random walk household sampling (see [11]). Adaptive/motivational potentials of D, i.e., the Disintegration-related tendency to irrationally assign causation, might lie in the fact that occasionally correct responses can carry a large fitness benefit in specific circumstances (see [11]).

## Disintegration and other PLEs models

It is not surprising that D model has little in common with Eysenck's Psychoticism factor: namely, in a recent study by Knežević et al. [12] meta-analytical correlation between Eysenck's Psychoticism dimension and various models of psychosis proneness was found to be only $r$ = .21. The similarity with Claridge's [27] four-factor model of schizotypy is not much higher: the only substantive overlap is with the Unusual Experiences factor. As already stated, Social Anhedonia—which is close to Claridge's Introvertive Anhedonia—was repeatedly found to be the primary indicator of low Extraversion [11]. Impulsive Non-conformity and the operationalization of Cognitive Distortions within Claridge's model were also outside the psychosis proneness domain—former, related to the domains of low Agreeableness and low Conscientiousness, and later to Neuroticism [12].

Concerning the four and five-factor models proposed to date (almost all of these models postulate the existence of Disorganization, Positive and Negative symptoms factors; some of them postulate additional Excitement/Mania and Depression factors [4,29,30], or Paranoia [31]), the D model differs primarily in the further specification of so-called positive and negative symptom factors. In the D model, the Positive Symptom factor is represented by the Perceptual Distortions, Paranoia, Magical Thinking, and Enhanced Awareness, while the Negative Symptom factor is represented by the Flattened Affect. The remaining three structures most frequently found in the five-factor models of psychosis proneness, i.e., Disorganization, Depression, and Mania are similar to the corresponding subdimensions of our model— General Cognitive/Executive Impairments, Apathy/Depression, and Mania.

D is also to a degree similar to the Psychoticism (P) factor of the PID-5 model of maladaptive traits [14], but also schizotypy/dissociation factors presented in the studies of Watson et al. [18] and Ashton & Lee [9]. However, these models are considerably narrower than D. They seem to emphasize some segments of this broad spectrum of strongly converging experiences/

behaviors constituting the domain according to the Disintegration, and many other, afore-mentioned models. For example, in the PID-5 model, P is conceptualized as a dimension of maladaptive personality, consisting of Unusual Beliefs and Experiences, Perceptual dysregulations, and Eccentricity. The absence of other aspects of psychosis proneness, such as Paranoia or Cognitive Disorganization (and the presence of more controversial Eccentricity facet—defined similarly to Ashton et al. [32], which is conceptually closer to O)–in the light of the available evidence–seems to require additional scrutinization at least. Namely, available evidence suggests that Paranoia and Disorganization should be conceptualized as important aspects of the domain, [6,33]. Although much broader than the constructs such as PID5 P, D is by no means similar to the General Factor of Psychopathology suggested by Caspi et al. [34]: the broad spectrum of anxiety-related, anhedonic, and antisocial phenomena is not a part of the D domain (apparently belonging to N, E, and A/C domains, or their maladaptive extensions: Emotional Instability, Detachment, and Antagonism/Disinhibition respectively).

### The affective valence of psychotic-like experiences

Most of PLEs are related to negative emotional states, and to lower positive emotions [33–36]. However, there is evidence that phenomena such as mania [37], spiritual, (mystical, numinous, peak experiences [38], out-of-body [39], and even some hallucinatory experiences [40] can be pleasurable. Therefore, we expect that the majority of PLEs are experienced as negative, but not all: phenomena constituting Mania, Enhanced Awareness, and perhaps Magical Thinking (actually MPLEs corresponding to these facets—grandiosity, all-embracing mind, and, perhaps, feeling the presence of strange forces around) are expected to be related to positive emotions, giving rise to the notion of the "happy schizotype" (e.g., [41]). This notion is important in the context of arguing against treating PLEs phenomena as exclusively causing distress and inner feeling of suffering.

### Can we conceptualize PLEs captured by inventories and questionnaires in non-clinical populations as manifestations of some of the HEXACO traits?

In this paper, we would like to test whether the well-established tendency that psychotic-like indices factorially separate from other personality traits could be an artifact of the methods that are usually used to assess them (retrospective self-reports or ratings). Namely, it was convincingly demonstrated that self-report of emotional experiences over short, versus long time frames, assess qualitatively different sources of self-knowledge. Thus, when reporting on emotions over short periods (e.g., last few hours) people retrieve from episodic emotion knowledge, but when reporting over long time frames (e.g., months or years), they retrieve semantic emotion knowledge [42]. The same authors showed that semantic emotion knowledge, unlike episodic one, can be influenced by the accessibility of emotion-related beliefs. It was in accordance with the findings of Barrett [43] showing that neurotics tend to retrospectively overestimate the degree to which they have experienced negative affect over a period of three months. It was likely due to accessibility of "individualistic / situation-independent belief" (i.e., "I am the kind of person who experiences a lot of negative affects") as labeled by Robinson and Clore [42]. It appears that the factors contributing to self-reports of PLEs retrospectively are likely explainable by the same accessibility model. Namely, it could be quite possible that general statements on PLEs (such as those in personality inventories), based on long time frames can be distorted similarly—by making them more consistent with these general self-beliefs. For example, people who are extremely open and creative are frequently described as unusual, odd, or eccentric by others, implying that they are different from "ordinary" or "normal" people.Ashton and Lee's [19] critique on the inappropriateness of labeling PLEs as "oddity" by

Watson et al. [18] -showing "oddity" to be a descriptor of O—testifies on this tendency to see a high level of O as having the flavor of something psychotic-like. Therefore, it is quite possible that many manifestations of high O, such as unconventionality and creativity—when retrospectively elaborated upon—tend to be more consistent with individualistic / situation-independent beliefs such as "I am the kind of person who tends to have a lot of unusual and weird experiences, thoughts and interests" or "I am a strange person". It might cause an artificially large variance of the PLE factor absorbing many aspects of O. When assessing these experiences in the short time frames (e.g., a couple of hours) the picture may change substantially: most of the momentary reports on PLEs, based on episodic knowledge that is not influenced by such biases, could be better predicted by O than by the inventory measures of PLEs. Therefore, even if the inventory items of PLEs tend to form a large factor separate from the other personality factors when jointly analyzed (as shown in e.g., [11,18,19,32]), it is theoretically possible that its existence might be, to a large extent, an artificial consequence of the retrospective biases that are known to systematically distort recollections of past experiences and events.

## Advantages of experience sampling methods

In the last 30 years, Experience Sampling Methods (ESM; [44]) or Ecological Momentary Assessment (EMA; [45]) became an important research tool in psychological assessment (e.g., [46,47]). The common characteristic of both ESM and EMA is the collection of self-reports or some indicators of behaviors, emotions, cognitions in real-time during the regular daily activities of the respondents. whilst traditional assessment methods, like self-reports, rely on retrospective reporting on one's personality which has many drawbacks, aforementioned recall biases amongst many others (such as socially desirable responding, acquiescent and extreme responding [48,49]). The ESM should be more robust to typical sources of error emerging in self-report assessment [45]. Measures based on ESM enable more direct and valid insight into the mental and behavioral phenomena. Thus, ESM studies yielded findings that challenged psychological theories, such as the result that higher anxiety about mathematics in girls usually reported in retrospective studies is not detected in everyday-life assessments [50], or that desire for sleep, sex, and leisure activity in everyday life is stronger than addictions to tobacco or alcohol [51]. The ESM/EMA proved to be a valid assessment method of outcomes related to mood dysregulation [52], anxiety [53], substance use disorders [54], and what is of special importance here, psychosis [55], and schizophrenia-spectrum traits [56]. It is also explicitly recommended as a novel and promising method in investigating psychotic phenotype [57].

One of the basic premises of this study is that assessing PLEs by ESM in a natural context of everyday life, twice per day, continuously for seven days, enables capturing the proneness to such behaviors/experiences in a way less distorted by retrospective recall biases. Although PLEs are mostly defined as related to the positive dimension of the psychotic phenotype [57], in this study their meaning is slightly different: MPLEs represent each of the nine facets postulated by the model of D (see section "Variables and instruments"). The main reason for the decision to select MPLEs that parallel facets of Disintegration is to investigate whether such a broader definition of D is warranted. Namely, it appears as self-evident that the core, perceptual distortions-related MPLEs (e.g., hallucinatory experiences assessed in [58]) would be predicted by D or PID-5 P rather than any of HEXACO traits. However, would MPLEs related to these less central contents of D be again predicted better by D or PID-5 P than HEXACO traits? Importantly, in the design of this study, we have MPLEs and personality assessment separated, both temporarily (one to two months) and contextually (experience sampling of everyday behavior in natural context vs. laboratory personality assessment).

## Goals of the study

The first goal of the study is to compare the power of trait-like dispositional sources of psychotic-like behaviors with the power of HEXACO traits to prospectively predict the nine MPLEs in the natural environment of everyday life. We hypothesize that these experiences are manifestations of a trait-like disposition toward PLEs, and—if such a disposition were included in the set of personality traits—it would be their primary predictor, while the predictive power of other traits would be weakly and sporadically present. If D score to a considerable extent reflects the aforementioned retrospective biases, it would not predict distant-in-time MPLEs, at least not better than HEXACO traits. If DeYoung et al. [15] are right in claiming that so-called positive psychotic-like symptoms are manifestations of high O, then MPLEs capturing perceptual distortions, magical thinking, and enhanced awareness should be predicted by this trait (more precisely by OE) equally or better than D.

Moreover, in line with the recently presented findings of this research group [21], the variance of OE shared with D and P should be the best predictor of MPLEs, not D or P only. As personality models such as Big Five postulate depression as a facet of Neuroticism (FFM), indices of apathy/depression should be better predicted by HEXACO E. The same could be postulated for somatization [59]. Some would also expect indices of heightened suspiciousness to be mapped onto negative Agreeableness, and mania to Extraversion [60]. Documenting that these MPLEs–less distorted by retrospective biases—are primarily predicted by the measures of dispositional tendencies toward PLEs, and not by other personality traits, would be further evidence of the validity of these dispositional measures.

The second goal of the study is to compare the power of three different conceptualizations of trait-like dispositional sources of PLEs. The first is D and the second is PID-5 P. As already stated, these two conceptualizations differ in scope—the former being broader than the latter. It could be that the narrower P disposition predicts MPLEs, which are not supposed to be aspects of the domain, better than any of the HEXACO traits. It would suggest that P might be too narrowly defined. It is also possible that P does not predict some MPLEs that are postulated to be aspects of the domain by the D model (i.e., those that cannot be defined as positive symptoms) better than other HEXACO traits. In this case, it would be plausible to conclude that the conceptualization of D might be overstretched. Finally, the third conceptualization of the dispositional source of PLEs is an amalgam of Psychoticism (here measured by DELTA and PID-5 P) and OE suggested by Blain et al. [21].

It seems that PLEs of the individuals not belonging to clinical populations are not exceptionally rare events (see [3,11,61,62]) which makes investigating everyday MPLEs important question. Thus, the third goal is to gain additional knowledge into the frequency, distribution, reliability, and intra—and inter-individual variations of the randomly, twice-per-day assessed MPLEs among university students. Although we expect the low frequency of MPLEs, and their distributions in a sample of university students to be strongly positively skewed—and phenomena such as depersonalization or feeling the presence of strange forces around to be rarest —we still expect that the presence of at least some of them characterizes the majority of our sample.

We also examine the within-individual relationships between affective states and PLEs (fourth goal). Related to this, the main advantage of ESM is the possibility to relate phenomena of interest and their daily changes with concomitant affective states. Previous studies show that ESM enabled a better understanding of the relationships between PLEs and affective states (e.g., [35]). Following these findings, we expect that all MPLEs are related to negative affects except those related to Mania, Enhanced Awareness, and possibly Magical Thinking–the last three we expect to be related to positive affective states. We have no specific expectations

regarding interactions between personality and affective states in MPLEs, therefore interactions between level-1 (affective states) and level-2 predictors (personality measures) will not be investigated.

To summarize, the main aim of the study is to further validate the construct of Disintegration. The special focus is on challenging the claim that less central contents of the D indeed constitute the domain–by diminishing the possibility that the strength of the claim critically depends on the retrospective biases present in the traditional inventory-based assessment of PLEs.

## Disclosures

All data are available on the Open Science Framework (https://osf.io/94t6p/). We report how we determined our sample size, all data exclusions, all manipulations, and all measures in the study. All procedures adhered to the principles of the Declaration of Helsinki.

## Method

### Sample

The sample consisted of 180 students of psychology, average age of 20.21 years (SD = 2.04), 83% female. Regarding the sample size in multilevel modeling, there is advice based on the experience, such as the recommendation that if hypotheses concern only relationships between a mean of a variable at level 1 and person-level (level 2) measure (such as a trait) 50 participants and 7 time-points should be adequate [63]. If there is an interest in the relationships between a within-person predictor and outcome (and there is no interest in cross-level interactions), 100 participants and 10 time-points should be adequate. Even if a researcher is interested in cross-level interactions (i.e., in Level-1 slopes)—which is not the case here—125 participants should suffice, who provide at least 14 data points. Our sample size is also far larger than the number found to lead to biased estimates of the second-level standard errors in MRCM (meaning a sample of 50 or less, [64]). Finally, using G*Power [65], a power analysis was conducted to detect a linear regression composite (consisting of 8 variables) that explains at least 9% of the variance in a dependent variable, using an alpha-level of 0.05 (two-tailed), with power set to 0.80. This power analysis revealed a required sample size of N = 175.

The Ethics committee of the Serbian Psychological Association at the Faculty of Philosophy, University of Belgrade, approved the study. Personality questionnaires were administered via the online Moodle platform in the laboratory, not allowing participants to skip the answers. The respondents filled in personality inventories during regular practicals, one month before experience sampling data were collected. Respondents received course credits for participation in the study which is a frequent practice in this type of study [63]. All participants signed the informed consent and voluntarily participated in the study.

### Variables and instruments

**Personality traits.** *HEXACO personality traits*. The HEXACO Personality Inventory-Revised (HEXACO PI–R; [66]) consists of 100 items with a 5-point Likert-type response scale ranging from 1 (strongly disagree) to 5 (strongly agree). The instrument assesses six basic personality traits and 24 facets: **H**onesty/Humility (facets: sincerity, fairness, greed avoidance, modesty), **E**motionality (facets: fearfulness, anxiety, dependence, sentimentality), e**X**traversion (facets: social self-esteem, social boldness, sociability, liveliness), **A**greeableness (facets: forgiveness, gentleness, flexibility, patience), **C**onscientiousness (facets: organization, diligence, perfectionism, prudence), and **O**penness (facets: aesthetic appreciation, inquisitiveness, creativity,

unconventionality). We used the Serbian form of HEXACO inventory [67]. Scores for personality domains and facets are calculated as average values of the scale items.

*Disintegration trait*. To assess the D trait, the DELTA questionnaire consisting of 110 items with a joint 5-point Likert type scale ranging from strongly disagree [1] to strongly agree [5] was administered [11]. DELTA enables the assessment of the nine aforementioned facets, while the total score serves as a measure of the D trait.

*DSM 5—Psychoticism domain*. To assess Psychoticism defined by the Alternative Model for Personality Disorders—AMPD [13], we used the Psychoticism scale from the Personality Inventory for DSM 5—PID-5 questionnaire [14]. This scale contains 33 items with a joint 4-point Likert-type scale (0 = very false or often false, to 3 = very true or often true). It assesses three facets of Psychoticism: Unusual Beliefs, Perceptual Distortions, and Eccentricity.

**Measures in ESM.** *Momentary psychotic-like experiences*. For the assessment of MPLEs, we created a 9-item questionnaire. These nine MPLEs parallel facets of D (to further investigate validity of MPLEs we suggest to analyze correlations between thought disorders diagnosis and items from 20-item Delta scale, close to MPLEs—data based on the national representative sample of Serbian population, https://osf.io/f8sje/). The item "My thoughts were confused" (confused thoughts) represents GC/EI facet; The item "I had a feeling that I was not myself" (depersonalization) represents PD; "I had a feeling that everyone was against me" (paranoid interpretation)—PAR; "I did not feel anything" (blunted affect)—FA; "My body did not function well" (dysfunctional body)—SOD; "I felt the presence of strange forces around me" (strange forces around)—MT; "I felt that my mind can overwhelm the entire world" (all-embracing mind)—EA; "I felt that everything is pointless" (meaninglessness)—A/DEP; "I felt like an almighty superman" (grandiosity)—MAN. The participants were asked to report to what extent the aforementioned experiences were present in the last two hours on a 5-point scale, from 0 = *does not describe my experience, or describes it just a little;* to 4 = *completely describes my experience.*

*Current mood*. For the assessment of positive and negative affect, we used the Positive and Negative Affect Schedule (PANAS; [68]). The PANAS assesses both positive and negative affect by asking participants to indicate to what extent they feel in the last 2 hours on a 5-point scale, from 0 = *not at all or very slightly*, to 4 = *extremely*. In this study, to assess self-reported affect, we have used the Serbian 10-item version of the PANAS scale [69]. The Serbian version of the PANAS showed adequate psychometric properties in previous studies (e.g., [69,70]).

## Procedure

**Experience sampling recordings** were collected using **the xSample—**a specially designed app developed and used for administering Experience Sampling Methodology ([71]). The software runs on the Android OS. In this study, respondents were notified via standard push notifications to fill in the survey, twice per day (signal contingent method). The time when respondents were beeped during each interval was semi-random: time periods were a) between 11 AM and 4 PM, and b) 5 PM and 11 PM (with randomly generated notifications within these time intervals). As part of the ESM, respondents completed the abovementioned questionnaires assessing the current mood and a questionnaire mapping MPLEs designed for the study. The respondents had to fill in the survey every day, during 7—but not necessarily—consecutive days. The software was programmed to continue sending notifications until 14 assessment points are collected. In the sample, 53% finished the study in 7 consecutive days, and an additional 30% in up to 10 days. The longest period to finish the study was 20 days (*N* = 3). All 180 students who agreed to participate finished the study, completing a total of 2520 recordings. Participants were instructed to complete the questionnaires based on their current

momentary thoughts and feelings ("*Over the last two hours I was. . .*"). They were compensated with credit points for participation in the study.

Personality inventories (HEXACO and DELTA) were filled-in 1–2 months before experience sample recordings. The inventories were administered to the participants by the authors of this article during the Individual Differences course practicum. Students were provided with feedback regarding their scores on the inventories.

### Analytic strategy

To investigate the amount of variance originating from the stable (interindividual differences) and the variance emanating from the varying factors, Multilevel Random Coefficient Modeling (MRCM), i.e., Hierarchical Linear Modeling (HLM) using HLM-6 software [72] was done. MRCM is a regression-based analysis appropriate for research designs where data are organized hierarchically, at more than one level. The two-level model was specified here (level 1, nine MPLEs, five negative and five positive affective states), and the respondent level (level 2, HEXACO traits + D or, HEXACO traits + PID-5 P, or HEXACO + variance common to OE, D, and PID-5 P). In other words, units of the analysis at level 1 were ESM recordings which were nested within aggregate units at level 2, i.e., respondents. The variance of each of the variables at level 1 was decomposed into within-individual and between-individual parts. ICC coefficients were calculated for each of the measures at level 1, as the ratio of between-individual variance and the overall variance. Means, *SD*s, reliabilities, skewness, and kurtosis were calculated for all variables. The outcomes were nine MPLEs and the mean MPLEs score. They were predicted by affective states at level 1 and by HEXACO + D, HEXACO + PID-5 P, or HEXACO + variance common to OE, D, and PID-5 P at level 2.

There are two important notes regarding the construction of the variables at level 2. The first one is related to the way the D domain score is calculated. To avoid the possibility of predictor—dependent variable overlapping, calculation of the total DELTA score in case of predicting a particular MPLE assumed the omission of the facet corresponding to that MPLE from the total DELTA score: e.g., in case of predicting Depersonalization, DELTA total score was calculated without taking into account PD facet score.

The second note regards the way the O score was calculated. Having in mind that proponents of the idea that PLEs are high levels of O insist on the fact that it is only OE that is essentially and positively related to PLEs—while I subcomponent correlates with O even negatively—we separated HEXACO O facets into OE (calculated as mean of Aesthetic Appreciation and Unconventionality) and I (calculated as the mean of Inquisitiveness and Creativity facets). We conducted three analyses with the personality scores as level 2 predictors: first, the MPLEs intercepts are predicted by HEXAC + OE + I + total DELTA score calculated as previously described, second, these intercepts are predicted by HEXAC + OE + I + PID-5 P domain score, and third, the intercepts are predicted by HEXAC + OE + I + variance common to OE, D, and PID-5 P enabling the comparison of the predictive power of the two measures of psychotic phenotype in the context of HEXACO predictors.

### Results

Descriptive statistics of the MPLEs and affects are presented in Table 1. While the distribution of MPLEs and negative affects were similarly positively skewed (lower scores predominate), the distribution of positive affects tended to be normal. Only ten subjects out of 180 always endorsed the alternative "*does not describe my experience, or describes it just a little*" when reporting on all of the nine MPLEs during the seven days of assessment. The rarest MPLEs was the presence of strange forces around him/herself (139 out of 180 subjects always endorsed

**Table 1. Descriptive statistics for the variables measured at level 1 (14 measurements aggregated across a subject, N = 180, 2508 records).**

| | Min | Max | M | SD | Sk | Ku | Rel | ICC |
|---|---|---|---|---|---|---|---|---|
| MPLEs total score | 0 | 2.26 | .37 | .39 | 1.15 | 4.71 | .96[a] | .67 |
| Confused thoughts | 0 | 4 | .85 | 1.12 | 3.45 | .18 | .92[a] | .46 |
| Depersonalization | 0 | 4 | .19 | .59 | 4.07 | 11.79 | .92[a] | .44 |
| Paranoid interpret. | 0 | 4 | .16 | .55 | 1.86 | 17.92 | .82[a] | .25 |
| Meaninglessness | 0 | 4 | .43 | .90 | 2.16 | 3.84 | .93[a] | .50 |
| Flattened Affect | 0 | 4 | .48 | .89 | 1.86 | 3.28 | .92[a] | .46 |
| Dysfunctional body | 0 | 4 | .31 | .78 | 2.70 | 6.72 | .90[a] | .40 |
| All-embracing mind | 0 | 4 | .45 | .92 | 2.06 | 3.35 | .95[a] | .60 |
| Strange forces around | 0 | 4 | .11 | .48 | 5.21 | 30.44 | .94[a] | .55 |
| Grandiosity | 0 | 4 | .39 | .85 | 2.37 | 5.06 | .94[a] | .53 |
| Negative affects | 0 | 1.93 | .45 | .39 | 1.23 | 1.86 | .89[a] | .43 |
| Positive affects | .61 | 3.74 | 1.95 | .85 | .54 | -.61 | .86[a] | .40 |
| Descriptive statistics for the domain scores from HEXACO and DELTA (level 2) | | | | | | | | |
| D | 1.28 | 3.82 | 2.20 | .46 | .59 | .24 | .96[b] | |
| H | 1.75 | 4.81 | 3.63 | .63 | -.45 | -.16 | .84[b] | |
| E | 1.69 | 4.94 | 3.45 | .62 | -.04 | -.21 | .83[b] | |
| X | 1.31 | 5.00 | 3.38 | .79 | -.42 | -.42 | .92[b] | |
| A | 1.19 | 4.81 | 3.05 | .66 | -.14 | .00 | .86[b] | |
| C | 2.06 | 4.94 | 3.70 | .65 | -.21 | -.63 | .87[b] | |
| O | 2.13 | 5.00 | 3.96 | .57 | -.68 | .63 | .83[b] | |
| OE | 2.13 | 5.00 | 4.04 | .58 | -.70 | .43 | .72[b] | |
| I | 1.75 | 5.00 | 3.89 | .66 | -.58 | .12 | .71[b] | |
| PID-5 P | .03 | 2.75 | .74 | .54 | 1.38 | 2.35 | .95[b] | |

Note: Min—lowest score; Max—highest score; Sk—skewness; Ku—kurtosis; Rel–reliability [a] MRCM reliability(MRCM reliability is calculated as: $\lambda = \tau_{00} / (\tau_{00} + \sigma/n_k)$; $\tau_{00}$ –variance of inter-individual differences; $\sigma$–variance of intra-individual differences; $n_k$–number of measurement points at level 1. This coefficient gives similar information as the percentage of the variance explained by the inter-individual differences, the difference consisting in the intra-individual differences here divided by the number of observations.), [b] Cronbach Alpha reliability; MPLEs–Momentary psychotic-like experiences; D—Disintegration; H—Honesty, E—Emotionality, X—Extraversion, A—Agreeableness, C—Conscientiousness; OE–Openness to experiences (Pure Openness, Aesthetic Appreciation and Unconventionality); I–Intellect (Inquisitiveness and Creativity); Theoretical range of the variables at level 1 are from 0 to 4, while for the variables at level 2 are 1–5 (except PID-5 P, which is 0–3).

"*does not describe my experience, or describes it just a little*"), whilst confused thoughts were the most frequent one (only 20 out of 180 subjects did not report having confused thoughts) during the seven days. Individual differences in MPLEs were substantial, generally larger than the intraindividual ones, as revealed in the ICC coefficients presented in Table 1. Intraindividual variations in MPLEs were lower than the variations of affects. To some extent, it can be ascribed to the skewness of the distribution of MPLEs but not entirely: distribution of negative emotional states was almost equally skewed, and still their intraindividual variability was higher than the variability of MPLEs. The most intra individually varying MPLE were paranoid interpretations (PAR), while the least varying one was the experience of all-embracing mind (EA). The precision of the estimation of the total MPLE score was high, .96 (see Table 1), pointing to the fact that 14 measurement points enable reliable measurement of MPLE phenomena. Nine MPLEs tend to converge, similarly as D facets tend to converge: Cronbach alpha for the total score based on the MPLEs aggregated within the persons was .84.

Descriptive statistics for variables at level 2 (personality variables) are displayed in Table 1. What distinguishes the population of students of psychology from the general population in Serbia, UK, or Germany [20], regarding the HEXACO, is the higher O (more than 1 *SD*).

**Table 2. Intercorrelations of personality traits assessed by HEXACO PI and DELTA inventory (N = 180).**

|  | H | E | X | A | C | O | OE | I | PID-5 P |
|---|---|---|---|---|---|---|---|---|---|
| D | -.10** | -.02 | -.40** | -.16** | -.27** | .29** | .26** | .26** | .85** |
| H |  | .14 | -.16 | .31** | .14** | -.06 | -.07 | -.05 | -.17* |
| E |  |  | -.04 | -.06 | .12 | -.11 | -.14 | -.07 | -.09 |
| X |  |  |  | .-.01 | -.06 | -.03 | -.04 | -.01 | -.29** |
| A |  |  |  |  | .04 | -.01 | .02 | -.03 | -.20** |
| C |  |  |  |  |  | -.01 | -.03 | .02 | -.25** |
| O |  |  |  |  |  |  | .90** | .92** | .33** |
| OE |  |  |  |  |  |  |  | .67** | .33** |
| I |  |  |  |  |  |  |  |  | .27** |

*Note*: D—Disintegration; H—Honesty, E—Emotionality, X—Extraversion, A—Agreeableness, C–Conscientiousness; OE–Openness to experiences (Pure Openness, Aesthetic Appreciation and Unconventionality); I–Intellect (Inquisitiveness and Creativity); PID-5 P–Psychoticism.

** $p < .01$.

* $p < .05$.

Compared to the sample representative for the general population in Serbia, they are approximately 0.5 *SD* lower on D [11].

Table 2 shows the intercorrelations of personality traits. The intercorrelations within HEXACO traits reflect the usual pattern [24]. D had the largest correlation with low X, while PID-5 P had the largest correlation with O. However, none of the correlations exceeded .40, except the correlations between the two measures of PLEs (D and PID-5 P) and the two aspects of O (OE and I). Despite the conspicuous difference in conceptualization and scope of D and PID-5 P, they share a fundamental amount of variance. The separation of O into OE and I yielded very similar low correlations with both measures of PLEs, i.e., D and PID-5 P.

## Analysis of the relations between variables at level 1

In the case of the MPLEs total score, both high negative and low positive affective states were found to be its reliable correlates (Table 3)—the relationships with the negative affects being stronger than the relationships with the low positive affects. The same pattern was detected in the case of depersonalization, dysfunctional body, meaninglessness, feeling the presence of strange forces around, and confused thoughts. In the case of all-embracing mind and grandiosity, it was the other way around—positive affective states were related to this experience, while negative states were related negatively, the later effects being much weaker than the former. Paranoid interpretations were related to negative affects only. Diminished affect was related to low positive affects, but was unrelated to negative affective experiences.

## Personality variables as predictors of MPLEs at level 2

The relationships between personality traits and MPLEs (intercepts) are presented in Table 3. All nine MPLEs were found to be predicted primarily by the D domain. Confused thoughts and meaninglessness were additionally predicted by X, independently from D. All-embracing mind and grandiosity were secondarily predicted by high X, low C, and low E. Diminished affect was predicted by low E and high A. Intellect, as an aspect of O separate from the OE aspect of O, was found to have a small contribution to the variance in feeling the presence of strange forces around. One should bear in mind that although the nine MPLEs were selected

**Table 3. Personality, i.e.HEXACO + (D or PID-5 P, or variance common to OE, D, and PID-5 P) and affective predictors of momentary psychotic-like experiences (each of the nine experiences and their mean score).**

| MPLEs | Mean of all nine MPLEs | | Confused thoughts | | Depersonal. | | Paranoid interpret. | | Meaningless. | | Flattened Affect | | Dysfunctional body | | All-embracing mind | | Strange forces around | | Grandiosity | |
|---|---|---|---|---|---|---|---|---|---|---|---|---|---|---|---|---|---|---|---|---|
| ICC | .67 | | .48 | | .44 | | .26 | | .50 | | .46 | | .40 | | .60 | | .57 | | .53 | |
| **Level 1** | B | t(179) | B | t(179) | B | t(179) | B | t(179) | B | t(179) | B | t(179) | B | t(179) | B | t(179) | B | t(179) | B | t(179) |
| Pos_aff. | -.04** | -4.81 | -.22** | -8.31 | -.06** | -3.82 | -.02 | -1.75 | -.11** | -5.94 | -.17** | -6.20 | -.08** | -3.89 | .14** | 6.22 | .00 | .35 | .18** | 7.43 |
| Neg_aff. | .18** | 10.24 | .66** | 13.00 | .17** | 4.93 | .27** | 7.74 | .41** | 9.66 | -.06 | -1.31 | .18** | 4.73 | -.08* | -2.60 | .07** | 2.65 | -.05 | -1.51 |
| **Level 2** | B | t(171) | B | t(171) | B | t(171) | B | t(171) | B | t(171) | B | t(171) | B | t(171) | B | t(171) | B | t(171) | B | t(171) |
| D | .53** | 8.48 | .78** | 5.93 | .34** | 4.29 | .25** | 4.87 | .47** | 4.19 | .60** | 5.93 | .44** | 4.81 | .74** | 5.75 | .26** | 3.76 | .59** | 5.23 |
| H | -.03 | -.76 | -.12 | -1.36 | -.04 | -.81 | -.02 | -.58 | .09 | 1.14 | .03 | .38 | -.05 | -.86 | -.10 | -1.12 | .01 | .13 | -.09 | -1.18 |
| E | -.06 | -1.52 | .11 | 1.33 | -.02 | -.33 | .00 | -.06 | -.05 | -.65 | -.22** | -3.26 | .08 | 1.37 | -.22** | -2.67 | -.07 | -1.71 | -.16* | -2.18 |
| X | .03 | .87 | -.17* | -2.43 | .02 | .37 | .00 | -.11 | -.18** | -3.02 | -.11 | -1.87 | -.03 | -.69 | .29** | 4.08 | .08 | 2.01 | .32** | 5.09 |
| A | .04 | 1.08 | .05 | .55 | .05 | .97 | -.03 | -.81 | .00 | -.02 | .14* | 2.06 | .09 | 1.46 | .06 | .80 | -.05 | -1.08 | .06 | .85 |
| C | .07 | 1.71 | .11 | 1.27 | .06 | 1.33 | .05 | 1.55 | .04 | .52 | -.06 | -.96 | -.02 | -.30 | .18* | 2.28 | .00 | .06 | .17* | 2.47 |
| OE | -.02 | -.41 | -.06 | -.47 | .05 | .74 | -.02 | -.46 | -.02 | -.22 | -.14 | -1.52 | .01 | .15 | .05 | .48 | .08 | 1.27 | -.13 | -1.27 |
| I | -.04 | -.9 | -.08 | -.73 | -.02 | -.32 | -.03 | -.73 | .06 | .62 | -.05 | -.61 | -.05 | -.76 | -.03 | -.26 | -.11* | -2.09 | -.03 | -.39 |
| P (PID-5) | .36** | 6.32 | .57** | 4.86 | .23** | 3.51 | .15** | 3.04 | .30** | 2.97 | .37** | 3.91 | .25** | 2.97 | .63** | 5.85 | .27** | 4.75 | .49** | 5.11 |
| H | -.03 | -.6 | -.11 | -1.19 | -.04 | -.78 | -.02 | -.49 | .09 | 1.07 | .02 | .33 | -.06 | -.88 | -.07 | -.82 | .02 | .45 | -.08 | -1.01 |
| E | -.04 | -.91 | .14 | 1.6 | .00 | .00 | .01 | .22 | -.03 | -.40 | -.17* | -2.36 | .09 | 1.50 | -.20* | -2.39 | -.06 | -1.42 | -.14* | -2.00 |
| X | -.02 | -.57 | -.23* | -3.17 | -.01 | -.34 | -.03 | -1.07 | -.21** | -3.33 | .17** | -2.81 | -.08 | -1.60 | .22** | 3.37 | .06 | 1.80 | .26** | 4.40 |
| A | .04 | .97 | .05 | .53 | .05 | .95 | .04 | -.92 | .00 | .00 | .12 | 1.77 | .08 | 1.26 | .07 | .89 | -.03 | -.80 | .07 | 1.02 |
| C | .04 | .93 | .08 | 1.01 | .05 | .97 | .03 | .88 | .01 | .11 | -.10 | -1.52 | -.05 | -.81 | .17* | 2.16 | .00 | .02 | .16* | 2.26 |
| OE | -.05 | -.84 | -.1 | -.78 | .03 | .41 | -.03 | -.51 | -.04 | -.34 | -.15 | -1.50 | .01 | .10 | -.05 | -.44 | .04 | .59 | -.18 | -1.73 |
| I | -.01 | -.15 | -.02 | -.18 | .00 | .06 | -.01 | -.21 | .09 | .96 | -.01 | -.73 | -.02 | -.31 | .00 | -.02 | -.09 | -1.82 | -.01 | -.08 |
| OE_D_P | .17** | 5.74 | .27** | 4.46 | .11** | 3.38 | .07** | 1.92 | .14** | 2.60 | .16** | 3.20 | .11** | 2.65 | .32** | 5.62 | .14** | 4.92 | .23** | 4.54 |
| H | -.03 | -.62 | -.11 | -1.19 | -.04 | -.82 | -.02 | -.57 | .08 | 1.03 | .02 | .0352 | -.06 | -.93 | -.07 | -.83 | .02 | .43 | -.08 | -.98 |
| E | -.03 | -.79 | .15 | 1.70 | .00 | .27 | .01 | .27 | -.03 | -.36 | -.16** | -2.19 | .09 | 1.50 | -.19* | -2.30 | -.06 | 1.44 | -.13 | -.78 |
| X | -.03 | -.89 | -.24** | -3.41 | -.02 | -.52 | -.04 | -1.18 | -.22** | -3.55 | -.18** | 3.05 | -.09 | -1.81 | .21** | 3.13 | .06 | 1.67 | .24** | 4.15 |
| A | .03 | .79 | .03 | .40 | .05 | .94 | -.04 | -1.01 | .00 | .10 | .11 | 1.53 | .07 | 1.20 | .06 | .80 | -.03 | -.76 | .06 | .79 |
| C | .03 | .72 | .07 | .86 | .04 | .86 | .03 | .90 | .00 | .01 | -.12 | -1.67 | -.06 | -.94 | .16* | 2.02 | .00 | .05 | .15* | 2.09 |
| I | -.03 | -.67 | -.06 | -.72 | .02 | .52 | -.02 | -.52 | .08 | 1.09 | -.08 | -1.22 | -.01 | -.17 | -.02 | -.25 | -.07 | -1.70 | -.09 | -1.35 |

*Note*: MPLEs—Momentary Psychotic-Like Experiences

** p < .01.

* p < .05. OE_D_P—Variance common to OE, D and PID-5 P. Total D score was calculated with the facet score—corresponding to a particular MPLE—excluded.

to represent the nine facets of the D model, each of them was predicted by the total D score from which the corresponding facet was excluded.

The same analysis was repeated but with PID-5 P instead of a D score. Importantly, PID-5 P was again the best predictor of all nine MPLEs, with a slightly lower predictive power in comparison to D. The structure of the other predictors contributing to MPLEs was highly similar to the one registered in the previous analyses. Finally, the analysis was repeated with the score containing variance common to OE, D, and P (OE score regressed onto D and P), instead of unique contributions from either D or P. The predictive power of such a score is considerable weaker compared to either D or P, and occasionally even weaker than HEXACO traits (in case of Blunted Emotion, Meaninglessness, and Grandiosity). Upon request of one of the reviewers we did the analyses with positive and negative emotions as dependent variables. In case of positive emotions, no matter whether HEXAC + OE + I + DELTA or HEXAC + OE + I + PID-5 P were predictor, the strongest ones were X and A. Negative emotions were best predicted by low X and D when HEXAC + OE + I + DELTA were predictors, and by low X and E when HEXAC + OE + I + PID-5 P were predictors (see S1 Table in Supplementary material at https://osf.io/94t6p/).

## Discussion

The most important finding is that all nine momentary, semi-randomly, twice-per-day measured psychotic-like experiences during one week were indeed better prospectively predicted by the D score than any of HEXACO traits—predictive contribution of HEXACO traits was small, sporadic, and unsystematic. Similar, although slightly weaker effects were obtained when D was replaced by the PID-5 P score. Therefore, MPLEs are the best prospectively predicted by the inventories designed to capture PLEs, not other personality measures. These results are based on highly reliable measures, both outcomes, and predictors. Importantly, the predictive findings on the relationships between D and MPLEs were not due to methodological overlap. Namely, the D score from the DELTA inventory was constructed with the facet—having content parallel to a particular MPLE–excluded, in each MRCM regression analysis. For example, the experience of meaninglessness was not predicted best by the D score due to D containing the corresponding facet of Apathy/Depression (as this facet was excluded from this particular regression analysis), but because the rest 8 facets of D substantively predicted the experience of meaninglessness (testifying on the coherence of the components of D). A similar finding with PID-5 P corroborates such a conclusion: PID-5 P does not contain a facet parallel to the experience of meaninglessness, and yet PID-5 score was the strongest predictor of this experience.

Within an individual, MPLEs were found to be more stable than affective states, both positive and negative, speaking against treating PLEs as transitory, fleeting experiences. Importantly, selected MPLEs were not so exceptionally rare even in the population of psychology students that can be considered pretty WEIRD (White/ Educated/ Industrialized/ Rich/ Democratic): just 6% of them constantly reported not having any of the nine MPLEs randomly assessed twice-per-day during the seven days. Actually, taking into account the nature of psychotic symptoms and empirical evidence on their low frequency even in the samples consisting of individuals diagnosed with psychosis [35,58], no one would expect that university students agree with the indicator items across the majority of measurement occasions. The distribution of MPLEs during the seven days was skewed, but this skewness was similar to the skewness of the distribution of negative affective experiences: respondents failed to strongly and uniformly disassociate themselves from the contents in these items. It appears that labeling MPLEs as

pathological phenomena—if it is based on their skewed distribution in non-clinical populations—is not more justified than qualifying negative affective states as pathological.

## MPLEs and personality traits

MPLEs based on the facets of the Disintegration model seem to primarily and overwhelmingly reflect a general proneness to PLEs. Depersonalization, paranoid interpretation, dysfunctional body, and strange forces present were predicted exclusively by D. The rest of the MPLEs were also predicted primarily by D, but additionally—although weakly—by X (all-embracing mind and grandiosity—positively, and confused thoughts and meaninglessness—negatively), E (all-embracing mind, grandiosity and flattened affect—negatively), and C (all-embracing mind and grandiosity—positively).

The important finding is that the narrower PID-5 P measure was also better in explaining variance in MPLEs based on the D model than HEXACO traits. Assuming that a) there is no other important dispositional tendency that can better explain MPLEs, and b) HEXACO inventory is a valid measure of the six traits purported to measure, it can be concluded that the phenomena not hypothesized to be the aspects of the PLEs domain by the PID-5 model actually should be considered as its parts. In other words, our findings indicate that the conceptualization of Psychoticism within the PID-5 model might be too narrow, omitting some important aspects of the proneness to PLEs.

Contrary to what can be expected based on DeYoung et al's arguments (2012) separation of OE from I did not increase its power of predicting MPLEs. If O is indeed a continuum underlying PLEs, one would expect to see something that we obtained with D or PID-5 P, i.e., OE being a primary predictor, or at least being a predictor of all or the majority of these experiences, or at least, MPLEs reflecting positive symptoms. Bearing in mind a recent contribution of this group of authors [21] it appears that they consider variance common to OE and psychotic phenotype—reflecting shared underlying mechanisms—as the most promising way to preserve Big Five taxonomy at the same time extending its predictive power to a vast realm of psychotic-like phenomena. As shown in Table 3, the score containing common variance of OE, P, and D has the incremental power to predict MPLEs over HEXACO in case of the most of MPLEs, but this predictive power is considerably lower compared with the unique contribution of either D or P. Of special notice is the low power of such a score to predict the core content of PLE–perceptual distortions. The pattern of these regression coefficients seems to reflect a simple but convincing tendency–critical in predicting MPLEs are properly defined psychotic-like dispositional tendencies, not dispositional tendencies captured by HEXACO traits. When such properly defined psychotic-like dispositional tendencies are diluted by a conceptual consideration–such as the one insisting that its variance should be shared with HEXACO OE–their predictive power regarding MPLEs is drastically weakened. Interestingly, even the experience of all-embracing mind—phenomenon belonging to the EA subdomain of D that usually has the secondary loading on O factor—was predicted exclusively by the measures of PLE, not OE, and considerably stronger by D or P compared to the score capturing commonality between OE, D, and P.

Grandiosity and experience of all-embracing mind were related to X and negatively to E. It seems that it is a consequence of the fact that these two MPLEs—unlike the rest of them–are related to positive affects, which are conceptualized as prominent aspects of Extraversion in the FFM model (as a separate subdimension), but also in HEXACO (as a strong constituent of Liveliness).

However, the fact that some of the investigated MPLEs were predicted exclusively by D or PID-5 P, and the others additionally—although weakly—by some HEXACO traits, suggests

that some of these MPLEs experiences might represent the core of the dimension and that the other might reflect more peripheral content of the trait. Constituting the periphery of the construct, its contiguity with other domains is, naturally, reflected in the correlations with these other domains. This is a likely explanation in the case of, for instance, grandiosity and meaninglessness. It is supported by the fact that D facet scales (MAN and A/DEP)—which these particular MPLEs correspond to—tend to have secondary loadings on the same personality traits (i.e., Extraversion and Neuroticism/Emotionality). Put differently, it seems that the correlations of these MPLEs with the other traits are not just a consequence of the selection of a particular indicator. In the case of confused thoughts, the fact that low X contributes to some extent to its variance we assume is a pure consequence of the selected indicator: in the majority of other indicators of GC/EI this contribution would likely not appear. In general, we assume that the selection of some other MPLEs from these nine aspects would result in very similar findings—the assumption worth further investigations.

We think that the possible interstitial nature of some well-known personality/clinical phenomena (the fact that they are induced by more than one trait) is something important to be aware of. The case of the experience highly saturated by negative emotions—meaninglessness (representing apathy/depression, A/D)—is a nice illustration. Usually, the A/D scale has secondary loadings on either Neuroticism/Emotionality [11] or low Extraversion. It seems that some aspects of A/D, such as those reflected in the sense of meaninglessness (but also in the loss of energy, fatigue, suicidality, sleep, or appetite dysfunctions), are more of disintegrative nature, while some other aspects, such as sadness and pessimism are more related to Neuroticism/Emotionality or low Extraversion. Instead of excluding such *no man's land* contents from the start of the quest into the structure of a personality trait, the field would benefit more if we carefully study them. Although factorially pure assessment tools are good for practical purposes, a full understanding of a personality trait comes not only from studying its core contents but also from investigating slightly peripheral ones. Thus, we think that these more emotional aspects of D, such as MAN and A/DEP—although tending to have secondary loadings on some other traits—are still important aspects of this latent dimension: they tell something important about D. One should bear in mind that they have as large loadings on the general D factor as other, more cognitive phenomena (such as paranoia and magical thinking). The precise charting of the domain is a prerequisite for a successful search for the endophenotypic models, neuroanatomical and neurochemical candidate mechanisms to explain individual differences in such a dimension. For example, it would be highly important to find out whether one of the neurological mechanisms suggested as an explanation for some cognitive impairments in psychosis (deficient representations of contextual information at the perceptual and cognitive level of functioning, [73]) can be applied not only to these cognitive dysfunctions but also to the specificities of the emotional processing of those who are psychotic or psychosis-prone. If not, we cannot claim that we have the neurobiological mechanism of the commonalities across the psychotic phenotype fully explained and understood.

It should be emphasized that the MPLEs in our study can be considered as milder indices of the psychotic continuum (in comparison to the psychotic symptoms from, for example, a cross-national study of the continuum of psychosis symptoms in the general population by Nuevo et al. [74]), thus having even more chances to be predicted by HEXACO. Despite this fact, they were only marginally and weakly predicted by HEXACO.

To conclude, the selected MPLEs are manifestations of the dispositional tendency beyond the HEXACO space. To have a comprehensive explanation of human behavior, which includes PLEs, it seems that the postulation of the existence of a Disintegration-like dispositional tendency is a reasonable and empirically well-sounded strategy [9–12,20].

## MPLEs and affective states

Our findings showed that emotional states are significant concomitants of MPLEs. Negative affective states correlated with the higher presence of the majority of MPLEs (which is in line with e.g., [35]), except for flattened affect, grandiosity, and an all-embracing mind. Positive emotional states were related to higher levels of grandiosity and the experience of an all-embracing mind, unrelated to paranoid interpretations, and negatively related to the remaining MPLEs. The last finding is in line with the findings on the existence of "happy schizotype" [41]. These phenomena comprise out-of-body-experience, and a wide range of phenomena that could be labeled as profound spiritual experiences, such as religious, peak, transcendental, mystical, numinous, psychedelic, ecstatic, or oceanic. The existence of these phenomena within the domain of PLEs further questions its exclusive qualifications as the domain of psychopathology.

## Limitations

Criticism can be directed towards the selection of the MPLEs and that they probably do not represent the highest registers of the continuum of severity of psychosis phenotype. Fonseca Pedrero and Debbane [57] emphasized the importance of differentiation between PLEs, schizotypal, subclinical psychotic symptoms, frank psychotic symptoms, and psychotic-spectrum disorders along the severity continuum. However, they admitted that "it is important to note that the boundaries between these phenomenological traits and experiences are fuzzy and sometimes unclear" ([57], p. 7). Finally, the selection of MPLEs from the medium and lower registers of the psychotic phenotype was an advantage in the context of the design of our study: as stated, it is probably not so challenging to show that the most severe psychotic symptoms cannot be predicted by the HEXACO or Big Five traits, the information is that even experiences/behaviors from the medium and lower level of this continuum cannot be adequately predicted by personality traits other than psychosis-proneness. Additionally, assessing the severest symptoms from the psychotic phenotype would not be an appropriate strategy if using a WEIRD sample such as students of psychology, known to have lower than average scores on the measures capturing proneness to PLEs.

The fact that the results of the study were based on a highly WEIRD sample (students of psychology) represents a limitation. As the presence of PLEs is expected to strongly influence many aspects of everyday adaptations [26], likely, the assessment of many easily available, highly functional segments of the general population (students, volunteers) would result in its skewed distributions, either because those with high D are unrepresented in such subpopulations or less willing to participate in the assessment or both. Therefore, if the study had been done using a more representative sample the findings would be likely even more in the direction of the main conclusions—especially those on not so exceptional rarity of PLEs and MPLEs, and their conceptualization as the normal personality variations. However, the range restriction especially in O scores (many respondents with higher-than-average scores), but also in D scores (many respondents with lower-than-average scores) could have decreased their correlation with MPLEs.

## Conclusions

MPLEs in a sample of university students were found to be rare, but not exceptionally rare: the frequency and skewness of their distributions were similar to the skewness of the distributions of negative emotional states. MPLEs as more straightforward and valid (i.e., not distorted by retrospective recall biases) experiential/behavioral indices of psychotic phenotype seem to have the meaning similar to dispositional measures of PLEs in personality inventories.

Namely, the domain of PLEs was the only consistent prospective dispositional predictor of this set of broadly sampled momentary experiences/behaviors. The predictive power of the other traits to MPLEs was small and unsystematic, i.e., conspicuously weak and sporadic in comparison to Disintegration to consider them major dispositional sources of these experiences/behaviors. Importantly, PID-5 P was also a better prospective predictor of the MPLEs than HEXACO traits. Considering the fact that MPLEs in this study were selected to parallel subdimensions of the D factor—which is considerably broader than PID-5 P–the last findings were interpreted as indicating a probability that some important aspects of the domain of PLEs were neglected by the PID-5 Psychoticism factor. Limiting the variance of psychotic-like predictors only to the one shared with OE considerably reduced its predictive power compared to HEXACO traits. It seems that a comprehensive explanation of psychotic-like behavioral and experiential regularities requires the postulation of a Disintegration-like dispositional tendency beyond HEXACO or the Big Five.

## Supporting information

**S1 Table.**
(DOCX)

## Author Contributions

**Conceptualization:** Goran Knežević, Ljiljana B. Lazarević.

**Data curation:** Goran Knežević, Aleksandar Zorić.

**Formal analysis:** Goran Knežević.

**Investigation:** Goran Knežević, Aleksandar Zorić.

**Methodology:** Goran Knežević, Aleksandar Zorić.

**Project administration:** Goran Knežević, Ljiljana B. Lazarević, Aleksandar Zorić.

**Software:** Ljiljana B. Lazarević.

**Supervision:** Goran Knežević.

**Validation:** Ljiljana B. Lazarević, Aleksandar Zorić.

**Writing – original draft:** Goran Knežević.

**Writing – review & editing:** Ljiljana B. Lazarević, Aleksandar Zorić.

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
