## [Decision Letter · Decision Letter 0]

1 Dec 2021

PONE-D-21-28042The meaning of momentary psychotic-like experiences in a non-clinical sample : a personality perspectivePLOS ONE

Dear Dr. Knezevic,

Thank you for submitting your manuscript to PLOS ONE. After careful consideration, we feel that it has merit but does not fully meet PLOS ONE’s publication criteria as it currently stands. Therefore, we invite you to submit a revised version of the manuscript that addresses the points raised during the review process.

We look forward to receiving your revised manuscript.

Kind regards,

Michelle Luciano

Academic Editor

PLOS ONE

Journal Requirements:

Additional Editor Comments:

As you will see both reviewers identified strengths with your research, but one reviewer was particularly concerned about the validity of the MPLE scale. In a revision, it will be essential to include further empirical work which establishes the validity, otherwise a very strong defense/discussion of it will be required (and may or may not be accepted by the reviewer). So you should specifically address their comments: "How, exactly, were these items arrived at? Would other items have worked better? Do they capture PLEs as assessed by psychosis diagnoses, clinical assessment, other validated measures, and the like?". In short, how can you be sure that the predictive patterns found aren't due to methodological overlap between scales?

Reviewers' comments:

Reviewer's Responses to Questions

**Comments to the Author**

1. Is the manuscript technically sound, and do the data support the conclusions?

Reviewer #1: Yes

Reviewer #2: Partly

2. Has the statistical analysis been performed appropriately and rigorously? 

Reviewer #1: Yes

Reviewer #2: Yes

3. Have the authors made all data underlying the findings in their manuscript fully available?

Reviewer #1: Yes

Reviewer #2: Yes

4. Is the manuscript presented in an intelligible fashion and written in standard English?

Reviewer #1: Yes

Reviewer #2: Yes

5. Review Comments to the Author

Reviewer #1: Review of PONE-D-21-28042, “The meaning of momentary psychotic-like experiences in a non-clinical sample: a personality perspective”

This manuscript reports on the associations of self-reports on the HEXACO personality factor scales, on the Disintegration (D) scale, and on the PID-5 Psychoticism scale with momentary psychotic-like experiences as reported through intermittent experience sampling on variables that correspond to the content of the Disintegration scale. The results, as obtained from a sample of Serbian psychology students, showed that momentary psychotic-like experiences were predicted strongly by the Disintegration scale and to a somewhat lesser extent by the PID-5 Psychoticism scale, but were not predicted by the HEXACO scales (apart from a few modest associations involving certain specific experiences), in spite of some plausible reason to expect high Openness to predict these experiences.

In my opinion, this manuscript is written competently and the research itself was undertaken competently. I found the results to be largely unsurprising but I can understand the authors’ explanation of the reasons for examining the issue empirically.

One minor suggestion for this manuscript is that it could be interesting to mention somewhere the associations of the positive affect and negative affect mood variables in relation to the HEXACO/Disintegration/Psychoticism characteristics.

I might mention also a couple of suggestions for future research:

First, it could be interesting to examine to what extent the with momentary psychotic-like experiences, and even the Disintegration variable itself, reflect individual differences in drug use. (It would be important to ensure that respondents are confident in their anonymity when asking such questions.)

Also, it could be useful to include some additional experience items that would be subjectively important but non-psychotic in content (e.g., difficulties or successes in work or other goals or in various kinds of relationships).

Some typographical errors to correct:

Cronbach’s name is misspelled as “Crobach” in at least one place

On page 21, “considerable” should be “considerably”

On page 24, “conceptual” is misspelled

On page 24, a parenthetical statement is opened by a parenthesis but the closing one is missing

Reviewer #2: Review of "The meaning of momentary psychotic-like experiences in a non-clinical sample : a personality perspective."

This paper examined the nature of MPLEs in terms of their prediction from mood, personality, and disposition to psychotic experiences. Using ESM, the authors demonstrated that MPLEs were most strongly predicted by disposition to psychotic experiences, much more so than normal-range personality variables as instantiated in the HEXACO.

This work has many strengths. The use of ESM with PLEs is relatively novel, and offers much in terms of clarifying the nature of PLEs as reported in self-report measures. Interpretation of PLEs has always been clouded somewhat by their relatively low base rates, and their seeming differences in meaning across different measurement formats; the use of ESM has the potential to offer clarity in this area by contextualizing them more thoroughly. The authors are also to be commended in refocusing attention on the overlap between normal range personality variables and psychosis. I think relationships between the domains of O (and other factors) and psychosis have generally been overstated or at least have never really been that well-resolved, and this paper helps draw attention to this.

My primary concern about the paper is that the measure of MPLEs seems to be somewhat underdeveloped, or lacking in broader validation. How, exactly, were these items arrived at? Would other items have worked better? Do they capture PLEs as assessed by psychosis diagnoses, clinical assessment, other validated measures, and the like? Currently it seems the authors attempted to mimic an existing measure of D in MPLEs; putting aside the question of how well this did or didn't work, it's maybe not surprising that the MPLEs then were best predicted by D.

Are the predictive patterns then due to methodological overlap, or more "substantive" factors? Currently it's a bit difficult to interpret the results. It seems clear that the D and PID-5 variables are better predictors than the HEXACO variables, but understanding why is more fraught with challenges.

On a more minor note, I think the manuscript could be streamlined quite a bit throughout. It is generally well-written but could be more succinct.

6. PLOS authors have the option to publish the peer review history of their article (what does this mean?). If published, this will include your full peer review and any attached files.

Reviewer #1: No

Reviewer #2: No

---

## [Author Response · Author response to Decision Letter 0]

25 Dec 2021

PONE-D-21-28042

The meaning of momentary psychotic-like experiences in a non-clinical sample: a personality perspective

PLOS ONE

Dear Dr. Knezevic,

Thank you for submitting your manuscript to PLOS ONE. After careful consideration, we feel that it has merit but does not fully meet PLOS ONE’s publication criteria as it currently stands. Therefore, we invite you to submit a revised version of the manuscript that addresses the points raised during the review process.

· A rebuttal letter that responds to each point raised by the academic editor and reviewer(s). You should upload this letter as a separate file labeled 'Response to Reviewers'.

· A marked-up copy of your manuscript that highlights changes made to the original version. You should upload this as a separate file labeled 'Revised Manuscript with Track Changes'.

· An unmarked version of your revised paper without tracked changes. You should upload this as a separate file labeled 'Manuscript'.

We look forward to receiving your revised manuscript.

Kind regards,

Michelle Luciano

Academic Editor

PLOS ONE

Dear Dr. Luciano, 

We are pleased to learn that our manuscript received what we consider a positive evaluation. We would like to thank you and the reviewers for their valuable comments. We have attentively addressed all comments and made revisions throughout the manuscript. All changes made in the text are visible as we use track-changes options. Our responses to all comments are given in blue letters, in italic. 

We hope our manuscript is now suitable for publication in your journal. 

Thank you once again for the time and effort invested in handling our manuscript. 

Kind regards, 

Goran Knežević

Additional Editor Comments:

As you will see both reviewers identified strengths with your research, but one reviewer was particularly concerned about the validity of the MPLE scale. In a revision, it will be essential to include further empirical work which establishes the validity, otherwise a very strong defense/discussion of it will be required (and may or may not be accepted by the reviewer). So you should specifically address their comments: "How, exactly, were these items arrived at? Would other items have worked better? Do they capture PLEs as assessed by psychosis diagnoses, clinical assessment, other validated measures, and the like?". In short, how can you be sure that the predictive patterns found aren't due to methodological overlap between scales?

Thank you for these thoughtful comments. We expanded the text to provide evidence on the quality and validity of selected items. A detailed response to the comment about the quality of MPLE items is given in responses given to Reviewer 2. 

Reviewers' comments:

Reviewer's Responses to Questions

Comments to the Author

1. Is the manuscript technically sound, and do the data support the conclusions?

Reviewer #1: Yes

Reviewer #2: Partly

2. Has the statistical analysis been performed appropriately and rigorously?

Reviewer #1: Yes

Reviewer #2: Yes

3. Have the authors made all data underlying the findings in their manuscript fully available?

Reviewer #1: Yes

Reviewer #2: Yes

4. Is the manuscript presented in an intelligible fashion and written in standard English?

Reviewer #1: Yes

Reviewer #2: Yes

5. Review Comments to the Author

Reviewer #1: Review of PONE-D-21-28042, “The meaning of momentary psychotic-like experiences in a non-clinical sample: a personality perspective”

This manuscript reports on the associations of self-reports on the HEXACO personality factor scales, on the Disintegration (D) scale, and on the PID-5 Psychoticism scale with momentary psychotic-like experiences as reported through intermittent experience sampling on variables that correspond to the content of the Disintegration scale. The results, as obtained from a sample of Serbian psychology students, showed that momentary psychotic-like experiences were predicted strongly by the Disintegration scale and to a somewhat lesser extent by the PID-5 Psychoticism scale, but were not predicted by the HEXACO scales (apart from a few modest associations involving certain specific experiences), in spite of some plausible reason to expect high Openness to predict these experiences. In my opinion, this manuscript is written competently and the research itself was undertaken competently. I found the results to be largely unsurprising but I can understand the authors’ explanation of the reasons for examining the issue empirically.

Thank you for the positive feedback. We are glad to hear that you consider our study interesting, relevant, and competently done. Although we completely agree with you that our results may not come as a big surprise to some scholars, the literature about the relationship between Openness and Psychotic-like experiences is still not unanimous, thus, we believe that our study may provide further clarification about the relationship between O and P. 

One minor suggestion for this manuscript is that it could be interesting to mention somewhere the associations of the positive affect and negative affect mood variables in relation to the HEXACO/Disintegration/Psychoticism characteristics.

Thank you for this comment. We have included a new Table in Supplement showing associations between affect and HEXACO/Disintegration/Psychoticism. We added a footnote to p. 22 directing the reader to the Table. 

I might mention also a couple of suggestions for future research:

First, it could be interesting to examine to what extent the with momentary psychotic-like experiences, and even the Disintegration variable itself, reflect individual differences in drug use. (It would be important to ensure that respondents are confident in their anonymity when asking such questions.)

Thank you for your thoughtful comments, and we hope that we have covered raised issues adequately. We completely agree with you that further investigation is needed. In line with that, we already started working on the relationship between Disintegration and different mental health problems, including drug/substance use. Recently we finished data collection for a large-scale project conducted on a representative sample of Serbian citizens, in which we conducted extensive measurement using the state of the instrument for the assessment of mental health - M.I.N.I.7.0.2. A research paper about the relevance of Disintegration for various mental health issues is soon to be submitted to a peer-reviewed journal. Protocol paper about the study is available, and we invite you to take a look: 

Maric, N.P. et al., (2021). Mental health in the second year of the COVID-19 pandemic: protocol for a nationally representative multilevel survey in Serbia. BMJ Open, http://dx.doi.org/10.1136/bmjopen-2021-053835

Also, it could be useful to include some additional experience items that would be subjectively important but non-psychotic in content (e.g., difficulties or successes in work or other goals or in various kinds of relationships).

Again, thank you for this suggestion. We are preparing a paper investigating the influence of situations on experiencing psychotic-like experiences and behaviors. 

We agree with you that further validation of the Disintegration trait is needed, and we plan a series of studies investigating the importance of Disintegration in various non-psychotic contents.

Some typographical errors to correct:

Thank you, we carefully read the manuscript again, and corrected several typos and errors. We apologize for this.

Cronbach’s name is misspelled as “Crobach” in at least one place

Thank you for noticing this. It has been corrected.

On page 21, “considerable” should be “considerably”

Thank you for noticing this. It has been corrected.

On page 24, “conceptual” is misspelled

Thank you for noticing this. It has been corrected.

On page 24, a parenthetical statement is opened by a parenthesis but the closing one is missing

Thank you for noticing this. It has been corrected.

Reviewer #2: Review of "The meaning of momentary psychotic-like experiences in a non-clinical sample: a personality perspective."

This paper examined the nature of MPLEs in terms of their prediction from mood, personality, and disposition to psychotic experiences. Using ESM, the authors demonstrated that MPLEs were most strongly predicted by disposition to psychotic experiences, much more so than normal-range personality variables as instantiated in the HEXACO. This work has many strengths. The use of ESM with PLEs is relatively novel, and offers much in terms of clarifying the nature of PLEs as reported in self-report measures. Interpretation of PLEs has always been clouded somewhat by their relatively low base rates, and their seeming differences in meaning across different measurement formats; the use of ESM has the potential to offer clarity in this area by contextualizing them more thoroughly. The authors are also to be commended in refocusing attention on the overlap between normal range personality variables and psychosis. I think relationships between the domains of O (and other factors) and psychosis have generally been overstated or at least have never really been that well-resolved, and this paper helps draw attention to this.

Thank you very much for your kind words. We appreciate your positive evaluation of the effort invested to run this study and the overall positive feedback on the quality of our study. We completely agree with you that the relationship between O domain and psychosis is generally overstated, but we still think it is important to provide further evidence on this relationship, as the literature is not unanimous regarding this issue. 

We would like to thank you for your thoughtful comments. We hope we addressed all your concerns adequately and that our manuscript is not improved. 

My primary concern about the paper is that the measure of MPLEs seems to be somewhat underdeveloped, or lacking in broader validation. How, exactly, were these items arrived at? Would other items have worked better? Do they capture PLEs as assessed by psychosis diagnoses, clinical assessment, other validated measures, and the like? Currently it seems the authors attempted to mimic an existing measure of D in MPLEs; putting aside the question of how well this did or didn't work, it's maybe not surprising that the MPLEs then were best predicted by D. Are the predictive patterns then due to methodological overlap, or more "substantive" factors? Currently it's a bit difficult to interpret the results. It seems clear that the D and PID-5 variables are better predictors than the HEXACO variables, but understanding why is more fraught with challenges.

A. We start with the reasons why our findings cannot be ascribed to methodological overlap – i.e. the fact that MPLEs parallel D sub-dimensions:

1. The following paragraph was already in the paper, but somewhat hidden in the Analytic strategy sections: “The first one is related to the way the D domain score is calculated. To avoid the possibility of predictor-dependent variable overlapping, calculation of the total DELTA score in case of predicting a particular MPLE assumed the omission of the facet corresponding to that MPLE from the total DELTA score: e.g., in case of predicting Depersonalization, DELTA total score was calculated without taking into account PD facet score” (p. 19). Obviously, this important point should be emphasized further and we did it in the first paragraph of the Discussion section. It reads: “Importantly, the predictive findings on the relationships between D and MPLEs were not due to methodological overlap. Namely, the D score from the DELTA inventory was constructed with the facet - having content parallel to a particularly MPLE – excluded, in each MRCM regression analysis. For example, the experience of meaninglessness was not predicted best by the D score due to D containing the corresponding facet of Apathy/Depression (as this facet was excluded from this particular regression analysis), but because the rest 8 facets of D substantively predicted the experience of meaninglessness (testifying on the coherence of the components of D). Similar finding with PID-5 P corroborates such a conclusion: PID-5 P does not contain the facet parallel to the experience of meaninglessness, and yet PID-5 score was the strongest predictor of this experience”.

2. To reiterate, we demonstrated that not only D predicted selected MPLEs better than HEXACO, it was also PID-5 P - the conceptualization of the PLE space that does not contain any facet accommodating for the experiences such as meaninglessness, grandiosity, or flattened affect. 

3. Validity of DELTA scales as measures of PLE: In the aforementioned recent study (cov2soul.rs, Maric et al., 2021) we related HEXACO + 20-item Disintegration scale to diagnoses established by M.I.N.I.7.0.2. The only significant personality predictor of psychotic disorders was the DELTA scale. A research paper about the relevance of Disintegration for various mental health issues is soon to be submitted to a peer-reviewed journal. Protocol paper about the study is available, and we invite you to take a look: 

Maric, N.P. et al., (2021). Mental health in the second year of the COVID-19 pandemic: protocol for a nationally representative multilevel survey in Serbia. BMJ Open, http://dx.doi.org/10.1136/bmjopen-2021-053835

4. In a recent study by Ristić et al. (2021, submitted for publication) the highest correlation with total PANSS (Positive and Negative Syndrome Scale, for rating the symptoms of schizophrenia) score was with the DELTA scale (higher than any of the Big Five traits measured by NEO PI-R) on N=161 patients with psychosis. 

The main reason for selecting MPLEs that parallel D facets was to demonstrate that momentary assessment of such experiences (devoid of retrospective biases) cannot be ascribed to anything else but proneness to psychotic-like experiences, i.e., the dispositional root beyond and above HEXACO. The D model was a suitable theoretical framework because it suggests a wider range of experiences to be treated as psychotic-like than some alternative models of PLE (such as PID-5). We could have selected experiences such as hallucinations (in terms of the D model, the most severe psychotic-like experiences are part of the D facets Perceptual Distortion and Somatoform Dysregulation), but it would have been far less convincing to demonstrate that such experiences were best predicted by D or PID-5 (for the moment ignoring the fact that such extreme experiences would be rare in the student population). Actually, a real epistemological gain is not to demonstrate that hallucinations belong to PLE space, but even experiences that were not a part of the most influential contemporary PLE models (roughly, experiences related to Apathy/Depression, Mania, Flattened Affect, General Cognitive/Executive Impairments, Paranoia, Somatoform Dysregulations). Fortunately, there is new and independent evidence that some of these neglected subdomains are part of the psychotic-like domain (thought disorders). For example, the recently appearing HiTOP model (Rugero et al., 2019) assumes that Paranoid and Manic phenomena belong to this domain.

On a more minor note, I think the manuscript could be streamlined quite a bit throughout. It is generally well-written but could be more succinct.

Yes, we absolutely agree with you. We did extensive editing throughout the text to improve the readability of the manuscript and corrected errors we noticed. We hope that the language is of much higher quality now.

---

## [Decision Letter · Decision Letter 1]

19 Jan 2022

PONE-D-21-28042R1The meaning of momentary psychotic-like experiences in a non-clinical sample : a personality perspectivePLOS ONE

Dear Dr. Knezevic,

Thank you for submitting your manuscript to PLOS ONE. After careful consideration, we feel that it has merit but does not fully meet PLOS ONE’s publication criteria as it currently stands. Therefore, we invite you to submit a revised version of the manuscript that addresses the points raised during the review process.

We look forward to receiving your revised manuscript.

Kind regards,

Michelle Luciano

Academic Editor

PLOS ONE

Additional Editor Comments:

You will see that reviewer 1 is satisfied with your revisions, but that reviewer 2 requires more information about the validity of the psychosis measure used. They have indicated reasonable options for you to take to make this clear and then they will be able to judge the merit of the scale.

Reviewers' comments:

Reviewer's Responses to Questions

**Comments to the Author**

1. If the authors have adequately addressed your comments raised in a previous round of review and you feel that this manuscript is now acceptable for publication, you may indicate that here to bypass the “Comments to the Author” section, enter your conflict of interest statement in the “Confidential to Editor” section, and submit your "Accept" recommendation.

Reviewer #1: All comments have been addressed

Reviewer #2: (No Response)

2. Is the manuscript technically sound, and do the data support the conclusions?

Reviewer #1: Yes

Reviewer #2: Partly

3. Has the statistical analysis been performed appropriately and rigorously? 

Reviewer #1: Yes

Reviewer #2: I Don't Know

4. Have the authors made all data underlying the findings in their manuscript fully available?

Reviewer #1: Yes

Reviewer #2: Yes

5. Is the manuscript presented in an intelligible fashion and written in standard English?

Reviewer #1: Yes

Reviewer #2: Yes

6. Review Comments to the Author

Reviewer #1: Review of PONE-D-21-28042R1, “The meaning of momentary psychotic-like experiences in a non-clinical sample: a personality perspective”

In my opinion this revised manuscript is ready for publication.

Reviewer #2: The authors have been thoughtful in replying to my previous concerns, and the manuscript is significantly improved. I appreciate their clarifications.

My remaining concern is about the validation of the psychosis measure being used. It seems they have some data pertinent to this, in a paper under review by Ristić et al. (2021). It is difficult to evaluate the manuscript without this; as such, it seems best to present those analyses in the current paper, or wait until the other paper has been accepted for publication, or to refer to a preprint that might be available. In the very least, citing and discussing the Ristić et al. (2021) paper in the form of a preprint seems appropriate. Otherwise it's difficult to interpret the psychosis measure being used.

7. PLOS authors have the option to publish the peer review history of their article (what does this mean?). If published, this will include your full peer review and any attached files.

Reviewer #1: No

Reviewer #2: No

---

## [Author Response · Author response to Decision Letter 1]

27 Feb 2022

The responses to reviewers are given in the file titled: letter_responses_esm_delta_after second_revision, which is attached

---

## [Decision Letter · Decision Letter 2]

1 Apr 2022

The meaning of momentary psychotic-like experiences in a non-clinical sample : a personality perspective

PONE-D-21-28042R2

Dear Dr. Knezevic,

We’re pleased to inform you that your manuscript has been judged scientifically suitable for publication and will be formally accepted for publication once it meets all outstanding technical requirements.

Kind regards,

Michelle Luciano

Academic Editor

PLOS ONE

Additional Editor Comments (optional):

You will see that reviewer 2 is now satisfied with your revision, so your paper can be accepted for publication.

Reviewers' comments:

Reviewer's Responses to Questions

**Comments to the Author**

1. If the authors have adequately addressed your comments raised in a previous round of review and you feel that this manuscript is now acceptable for publication, you may indicate that here to bypass the “Comments to the Author” section, enter your conflict of interest statement in the “Confidential to Editor” section, and submit your "Accept" recommendation.

Reviewer #2: All comments have been addressed

2. Is the manuscript technically sound, and do the data support the conclusions?

Reviewer #2: Yes

3. Has the statistical analysis been performed appropriately and rigorously? 

Reviewer #2: Yes

4. Have the authors made all data underlying the findings in their manuscript fully available?

Reviewer #2: Yes

5. Is the manuscript presented in an intelligible fashion and written in standard English?

Reviewer #2: Yes

6. Review Comments to the Author

Reviewer #2: The authors have addressed my concerns. I appreciate their openness.

7. PLOS authors have the option to publish the peer review history of their article (what does this mean?). If published, this will include your full peer review and any attached files.

Reviewer #2: No

---

## [Editor Report · Acceptance letter]

6 Apr 2022

PONE-D-21-28042R2 

The meaning of momentary psychotic-like experiences in a non-clinical sample: a personality perspective 

Dear Dr. Knežević:

I'm pleased to inform you that your manuscript has been deemed suitable for publication in PLOS ONE. Congratulations! Your manuscript is now with our production department. 

Kind regards, 

on behalf of

Dr. Michelle Luciano 

Academic Editor

PLOS ONE